# Diversity, Distribution, and Ecology of Fungi in the Seasonal Snow of Antarctica

**DOI:** 10.3390/microorganisms7100445

**Published:** 2019-10-12

**Authors:** Graciéle C.A. de Menezes, Soraya S. Amorim, Vívian N. Gonçalves, Valéria M. Godinho, Jefferson C. Simões, Carlos A. Rosa, Luiz H. Rosa

**Affiliations:** 1Departamento de Microbiologia, Instituto de Ciências Biológicas, Universidade Federal de Minas Gerais, Belo Horizonte 31270-901, Brazilsorayasander@gmail.com (S.S.A.); viviannicolau@yahoo.com.br (V.N.G.); valeriagods@gmail.com (V.M.G.); carlrosa@icb.ufmg.br (C.A.R.); 2Centro Polar e Climático, Universidade Federal do Rio Grande do Sul, Porto Alegre 91201-970, Brazil; jefferson.simoes@ufrgs.br

**Keywords:** Antarctica, ecology, fungi, snow

## Abstract

We characterized the fungal community found in the winter seasonal snow of the Antarctic Peninsula. From the samples of snow, 234 fungal isolates were obtained and could be assigned to 51 taxa of 26 genera. Eleven yeast species displayed the highest densities; among them, *Phenoliferia glacialis* showed a broad distribution and was detected at all sites that were sampled. Fungi known to be opportunistic in humans were subjected to antifungal minimal inhibition concentration. *Debaryomyces hansenii*, *Rhodotorula mucilaginosa*, *Penicillium chrysogenum*, *Penicillium* sp. 3, and *Penicillium* sp. 4 displayed resistance against the antifungals benomyl and fluconazole. Among them, *R. mucilaginosa* isolates were able to grow at 37 °C. Our results show that the winter seasonal snow of the Antarctic Peninsula contains a diverse fungal community dominated by cosmopolitan ubiquitous fungal species previously found in tropical, temperate, and polar ecosystems. The high densities of these cosmopolitan fungi suggest that they could be present in the air that arrives at the Antarctic Peninsula by air masses from outside Antarctica. Additionally, we detected environmental fungal isolates that were resistant to agricultural and clinical antifungals and able to grow at 37 °C. Further studies will be needed to characterize the virulence potential of these fungi in humans and animals.

## 1. Introduction

Antarctica is composed of special ecosystems that allow for the study of the taxonomy and ecology of resident life forms under extreme conditions [1]. Among the eukaryote organisms living in Antarctica, fungi have received special attention due their ability to colonize and survive in different Antarctic habitats/substrates. Most studies on fungi in Antarctica have focused on fungi living in terrestrial and marine environments, such as lakes [2,3], soils [4,5], historic woods [6], plants [7,8], and macroalgae [9,10]. In these Antarctic habitats, fungi are represented mostly by the taxa of the phyla *Ascomycota* and its anamorphs, followed by *Basidiomycota*, *Mortierellomycota*, and *Chytridiomycota* [11,12].

Snow habitats are ecosystems characterized by permanently low temperatures, low nutrient composition, and high levels of ultraviolet radiation at high altitudes [13]. Snow represents an important component of the cryosphere and occupies about 35% of the Earth’s surface during the Northern Hemisphere winter [14], covering over 99% of the 13.8 million km^2^ of the Antarctic continent. Antarctic seasonal snow, that is the snow deposited in the last winter season, can be considered a cryptic microhabitat for microorganisms directly related to the atmosphere, the precipitation of dust, and microbial cells, as well as other organic materials suspended in the air [14].

According to Kang et al. [15], falling snow greatly contributes to the deposition of airborne contaminants via the removal of atmospheric aerosols and the absorption of gas phase vapors. Industrial and agricultural activities have a great impact on these natural habitats due the dispersion of greenhouse gases, nitrogen compounds, and other synthetic chemicals, which, when released in high amounts, are dispersed in the atmosphere and transported by winds to remote areas of the planet, including the polar regions [16]. Previous studies have already demonstrated that several pesticides and semi-volatile organic compounds circulate and accumulate in remote ecosystems across the world, including high-elevation and high-latitude habitats [17,18,19]. Kang et al. [15] detected 22 persistent organic pollutants (POPs), identified as organochlorine pesticides (OCPs), in snow in Antarctica; among them, α-hexachlorocyclohexane (HCH), γ-HCH, and hexachlorobenzene (HCB) were frequently found in snow at different concentration ranges. The Antarctic Peninsula is close to South America, which is known for its intense agricultural and industrial activities and which employs the use of chemicals to combat plant and animal pests and diseases. These chemicals are then transported by winds to the Antarctic atmosphere to be then deposited in the snow by precipitation [16].

Though studies on the fungi found in snow in Antarctica are scarce, the presence of cold-tolerant cosmopolitan and psychrophilic endemic fungi living at the edge of life has been demonstrated [20,21,22,23,24], which are in contact with different compounds transported by the air and deposited in the snow, including pollutants with chemical structures similar to agricultural and clinical antifungal drugs. These pollutants may work as selective antimicrobial agents on Antarctic fungi living in snow. In this context, the present study aimed to characterize the fungal diversity present in the seasonal snow of the Antarctic Peninsula islands and investigate the possible resistance of species and strains against agricultural and clinical antifungal drugs.

## 2. Methods

### 2.1. Snow Sampling, Concentration, and Fungal Isolation

Seasonal snow samples (snow deposited during the last winter season) were collected in six different regions of the Antarctic Peninsula and in the South Shetland Islands between November and December 2015 (Appendix A). Approximately 10 kg of snow from the top, middle, and base layers (10 cm above soil) were collected in triplicate using a sterile plastic scoop, and these were stored in sterilized 20 L plastic bags (Appendix A). The samples were thawed in a microbiology laboratory on board the Brazilian polar ship Admiral Maximiano over a period of 12 h. After snow thaw, a mix of 10 L of water were obtained, and the physicochemical properties of the melted snow sample were measured using the Hanna multi-parameter probe HI 9828 (manufacture Hanna instruments, Woonsocket, RI, USA). A total of 1.5 L of melted snow was filtered through a membrane with a diameter of 47 mm (Millipore, Billerica, MA, USA) in triplicate experiments. The membranes were inoculated on a Sabouraud agar (Acumedia Lab, Lansing, MI, USA) and on a minimal Medium (6.98 g K_2_HPO_4_; 5.44 g KH_2_PO_4_; 1.0 g (NH_4_)_2_SO_4_; 1.1 g MgSO_4_ 7H_2_O, 0.25 g peptone, 20 g agar per L, pH 6.8 ± 0.2), containing 200 µg mL^−1^ of chloramphenicol (Sigma-Aldrich, Saint Louis, MO, USA), and these membranes were incubated at 10 °C for 30 days. The fungi were purified in fresh Petri dishes containing the Sabouraud agar and deposited in the Collection of Microorganisms and Cells of the Universidade Federal de Minas Gerais, Brazil, under the code UFMGCB in cryotubes at −80 °C and in distilled sterilized water [25] at room temperature.

### 2.2. Fungal Identification

The protocol for the DNA extraction of the cultivable fungi was conducted according to that procedure of Rosa et al. [1]. For the filamentous fungi, the internal transcribed spacer (ITS) region was amplified using ITS1 and ITS4 primers [26], which were amplified as described by Rosa et al. [1]. The amplification of the β-tubulin [27] and ribosomal polymerase II genes (*RPB2*) [28] were utilized for fungal taxa with low intraspecific variation according to protocols established by Gonçalves et al. [29]. The yeasts were identified according to methods proposed by Kurtzman et al. [30]. Yeast molecular identities were confirmed as described by Lachance et al. [31]. The fungal sequences were analyzed with the program MEGA7 [32]. All sequences were compared with type and/or annotated fungal sequences deposited in the GenBank database using the Basic Local Alignment Search Tool (BLAST at http://www.ncbi.nlm.nih.gov) [33]. Representative consensus sequences of the fungal taxa were deposited into the GenBank database (Table 1). To achieve species-rank identification based on ITS, β-tubulin data, and ribosomal polymerase B2, the consensus sequence was aligned with all sequences from related species retrieved from the NCBI GenBank database using BLAST [34]. The information about fungal classification generally followed the databases of dictionary Kirk et al. [35], the website MycoBank (http://www.mycobank.org) [36], and the Index Fungorum (http://www.indexfungorum.org) [37].

### 2.3. Morphological Characterization

After sequence analysis, *Penicillium* species with unknown taxonomic positions were characterized according to their macroscopic properties (colony color, texture, reverse color, border type, sporulation, and the production of yellow soluble pigments). The diameters of the colonies were observed on yeast extract agar (YES, yeast extract 3 g, peptone 5 g, agar 20 g per L) and Czapek yeast extract agar (CYA, Himedia, Mumbai, India) media at 25 and 30 °C, respectively, for 14 days [38]. The resulting colors were graded according to the specifications proposed by Kornerup and Wanscher [39].

### 2.4. Diversity, Richness, Dominance, and Distribution

The diversity, richness, and evenness indices of (i) Fisher’s α, (ii) Margalef’s, and (iii) Simpson’s, respectively, were calculated to estimate the fungal assembles diversity in Antarctic snow. In addition, the distribution among the fungal assemblages were determinates by Sorensen (QS) and Bray-Curtis (B) indices. All indices were calculated with 95% confidence and a bootstrap of 1.000 iterations using the program PAST, version 1.90 [40]. Additionally, a Venn diagram was produced according to the work of Bardou et al. [41] to illustrate the distribution of the fungal assemblages of the seasonal snow Antarctic.

### 2.5. Fungal Growth Responses to Temperature

Fungi known to be opportunistic in humans were grown on a Sabouraud agar at 10, 15, 20, 25, 30, 35, and 37 °C on Petri dishes to determine their growth profile at different temperatures. The filamentous fungi were inoculated by transferring blocks (4 mm^2^) from 10-day-old pre-cultures grown at 10 °C. Microtiter plates were incubated in triplicate for 7 days. Yeast was cultured on YM for 72 h at 10 °C. Then, 10 µL of cells with a turbidity of 0.5 on the MacFarland scale (1.5 × 10^6^ cells mL^−1^ at 70% transmittance) were inoculated at the center of the Petri dishes containing Sabouraud media. All plates were incubated in triplicate for 14 days to detect the presence or absence of growth.

### 2.6. Minimum Inhibitory Concentration (MIC) of Antifungal Drugs Benomyl, Fluconazole, and Amphotericin B

Fungi known to be opportunistic in humans were subjected to minimum inhibitory concentration (MIC) determination for the antifungal drugs benomyl (Sigma, Steinheim, North Rhine-Westphalia, Germany) and fluconazole (Sigma, Saint Louis, MO, USA). Isolates with an MIC ≥ 64 µg/mL for fluconazole that also exhibited growth at 37 °C were subjected to MIC determination for the antifungal drug amphotericin B (Sigma, Saint Louis, MO, USA). The MIC protocol was performed using a modified version of the method used for yeast [42] and filamentous fungi [43] in a Roswell Park Memorial Institute (RPMI-1640) medium (INLAB, Diadema, SP, Brazil) using 96-well microtiter plates. The yeast cells and fungal spores were treated with benomyl concentrations ranging from 0.078125 to 40 µg mL^−1^ [44], fluconazole from 0.062 to 64 µg mL^−1^, and amphotericin B from 0.007 to 8 µg mL^−1^. Microtiter plates containing spores and yeast cells were incubated at 25 °C for 48 h for benomyl and fluconazole and at 37 °C for 48 h for amphotericin B. All MIC assays were performed in duplicate.

## 3. Results

### 3.1. Snow Fungal Community

From the different snow samples, 234 fungal isolates were obtained and identified as 51 taxa of 26 genera (Table 1). Among the yeast species identified, *Bannozyma yamatoana*, *Cystobasidium pallidum*, *Glaciozyma antarctica*, *Leucosporidium fragarium*, *Leucosporidium golubevii*, *Phenoliferia glacialis*, *Phenoliferia psychrophenolica*, *Phaeococcomyces* sp., *Rhodotorula mucilaginosa*, *Hamamotoa singularis*, and *Vishniacozyma victoriae* displayed the highest densities (≤300 CFU L^−1^). In contrast, filamentous fungi taxa showed the lowest density values. Twenty-nine fungi (56.8%) occurred in low densities (≤50 CFU L^−1^); these represented singletons and were considered rare taxa within the seasonal snow fungal community. Twenty-two taxa showed low molecular similarities or inconclusive taxonomic definitions when compared with the sequences of known fungi deposited in the GenBank database. Among them, *Bionectriaceae* sp., *Cystobasidium* sp., *Genolevuria* sp., *Helotiales* sp., *Galactomyces* sp., and *Hyaloscyphaceae* sp. showed the lowest query coverage and identity percentages, and these may represent new species.

### 3.2. Diversity and Distribution

The seasonal snow fungal community showed high values of diversity (Fisher α), richness (Margalef’s), and dominance (Simpson’s) (Table 2). The highest ecological indices were found in the snow of King George Island, followed by Snow Island, Trinity Island, Deception Island, Robert Island, and the Arctowski Peninsula. In addition, the similarities between the fungal assemblages of the different snow segments of each site showed that, except for those of King George Island, Segments 2 and 3 were more similar than Segment 1 (Appendix A). Eleven yeast species displayed the highest densities; among them, *Phenoliferia glacialis* showed a broad distribution and was detected at all sites that were sampled (Appendix A).

### 3.3. Opportunistic Virulence Potential

Five isolates of *R. mucilaginosa*, one of *P. chrysogenum*, *Penicillium* sp. 3, and *Penicillium* sp. 4 showed capacity for resistance to the clinical antifungal drug fluconazole at 25 °C. Two isolates of *Debaryomyces hansenii* were resistant to benomyl but susceptible to fluconazole (Table 3). In addition, isolates of *R. mucilagina* displayed MIC values of 0.03 and 0.125 µg mL^−1^ against amphotericin B at 37 °C (Table 3). However, we did not detect a clear correlation between the antifungal benomyl, used in agriculture, with the cross-resistance of fungi against fluconazole and amphotericin B, used in clinical therapy.

## 4. Discussion

### 4.1. Taxonomy, Diversity, and Distribution

In the winter seasonal snow of the Antarctic Peninsula, we detected a rich fungal community. Within this community, the yeasts *B. yamatoana*, *C. pallidum*, *G. antarctica*, *L. fragarium*, *L. golubevii*, *P. glacialis*, *P. psychrophenolica*, *R. mucilaginosa*, *H. singularis*, and *V. victoriae*, all of which are known to occur in tropical, temperate, and polar environments, were found at high densities. Generally, in extreme environments, fungal assemblages are dominated by *Ascomycota*.

*Rhodotorula* includes pigmented yeast species that are found worldwide and can abundantly grow in extreme environments [45,46]. *Rhodotorula mucilaginosa* is ubiquitous and is present in different habitats and substrates, including cold and extreme environments [47]. In Antarctica, *R. mucilaginosa* is commonly isolated from terrestrial and marine substrates [48,49]. Despite the extensive snow cover of the Antarctic continent, our study represents the first report on the high density of *R. muscilagionsa* in snow samples of different sites in Antarctica.

Of all 51 taxa found in the Antarctic snow, only *Phenoliferia glacialis* (a synonym of *Rhodotorula glacialis*) was isolated from all sites sampled in this work and with high densities. *Phenoliferia glacialis* is a psychrophilic yeast that exhibits high metabolic versatility, and together with *P. psychrophenolica* (*Rhodotorula psychrophenolica*), was originally obtained from the Etendard Glacier in France and the Stubaier Glacier and cryoconite in Austria [46]. Recently, Ferreira et al. [50] isolated *P. glacialis* and *P. psychrophenolica* from the leaves of *Deschampsia antarctica*, mosses, and different biofilms present in the Antarctic Peninsula. *Cystobasidium pallidum* (*Rhodotorula pallida*) was obtained from different cold environments, such as the rhizosphere in Russia [51] and the seawater of the Indian Ocean [52]. In Antarctica, *C. pallidum* was detected in snow, rocks, and the Ross Dependency in the Antarctic Peninsula [53,54].

*Vishniacozyma victoriae* (a synonym of *Cryptococcus victoriae*) was first described in Antarctica [55] but was subsequently detected in penguin guano, soil, sediment, freshwater samples, the rhizosphere of *D. antarctica* [48], macroalgae [10], lichens [56], and as endophytes of *D. antarctica* and *Colobanthus quitensis* [57]. In addition, V. victoriae was detected in environments outside of Antarctica, such as soil and rhizosphere soil in Korea [58], seawater in Portugal [59], the gut of the insect *Chrysoperla rufilabris* in the USA [60], an industrial malting area and indoor air in Finland [61,62], in glacial ice from the Arctic [63], in a dry meat processing factory in Norway [64], and in the Italian Alps [65,66]. Garcia et al. [67] proposed that *V. victoriae* represents a generalist species with the ability to tolerate different stressful environments. Antony et al. [22] previously reported *V. victoriae* in the snow of Victoria Land, Antarctica.

The *Glaciozyma antarctica* species represents a re-classification of *Leucosporidium antarcticum*, originally described by Turchetti et al. [68]. In Antarctica, *G. antarctica* was previously isolated from Antarctic marine waters [69], soils around to Lake Fletcher, Lichen and Taylor Valleys, and on dead sponge [68], seawater of the Weddell Sea near Joinville Island, in the soils of South Victoria Land and Lake Fletcher [66], and in the seawater of the Northern Antarctic Peninsula [70].

*Leucosporidium fragarium* was obtained from the melt water river of the Frias Glacier of Mount Tronador [71,72]. *Leucosporidium golubevii* was isolated from the river water in the north of Portugal [73] and as endophytic fungus present in leaves of *C. quitensis* [57]. *Bannozyma yamatoana* (*Bensingtonia*
*yamatoana*) was reported in Antarctica, present in soil and ornithogenic soil [48], and associated with thalli of lichen *Usnea antarctica* [56].

Generally, in extreme environments fungal communities are dominated by *Ascomycota* filamentous fungi [74]. However, our results demonstrate that the basidiomycetous yeast assemblage was the dominant form in the winter seasonal snow of Antarctica. Similar results were obtained by de García et al. [71] from yeasts isolated from glacial melt water rivers originating from glaciers in the Argentinean Patagonia. Butinar et al. [62] detected viable yeasts in the ice layers of Arctic glaciers in the Svalbard Islands (Norway), and Turchetti et al. [65] detected viable yeasts in Alpine glaciers; however, in both studies, the yeasts occurred at low densities.

### 4.2. Resistance against Antifungal Drugs at 37 °C

We screened fungi previously reported as potential pathogenic taxa against the antifungal compounds benomyl (used as a pesticide in agriculture), fluconazole, and amphotericin B (used in medicine), which have structures composed of aromatic rings and which are similar to some POPs and OCPs previously found in snow [15]. *Debaromyces hansenii*, *R. mucilaginosa*, *Penicillium*
*chrysogenum*, and *Penicillium* sp. 3 and sp. 4 showed resistance against the antifungal drugs tested.

*Debaryomyces hansenii* is considered a cosmopolitan yeast species that is able to occupy and colonize different ecological niches and ecosystems. In Antarctica, *D. hansenni* was previously recovered from the rhizosphere of *D. antarc*tica and soil [48], as well as the soil and thalli of lichens [75], macroalgae [76], freshwater, and marine water [77]. However, *D. hansenii* (also reported as *Candida famata*, its anamorph) has been previously found to cause opportunistic human infections [78,79,80]. Moreover, there have been reports of higher MIC values of fluconazole for *D. hansenii* than for pathogenic yeasts species [81]. Though several isolates found in the snow show susceptibility to fluconazole and are unable to grow at 37 °C, *D. hansenii* is able to synthesize or tolerate several toxins and has been reported to show resistance against several environmental pesticides, among them benomyl [81]. Similarly, we found isolates of *D. hansenii* that were resistant to benomyl.

*Rhodotorula mucilaginosa* has recently emerged as an opportunistic fungal pathogen. *Rhodotorula mucilaginosa* is able to cause outbreaks of skin infections in chickens and lung infections in sheep [82,83], and it is able to worsen the condition of immunosuppressed patients [84]. Here, *Rhodotorula mucilaginosa* present in snow in Antarctica was found to be resistant to fluconazole and susceptible to amphotericin B. Similar results were reported by Spiliopoulou et al. [85], who found that opportunistic *Rhodotorula* species was resistant to fluconazole but susceptible to echinocandins and amphotericin B. Generally, opportunistic *R. mucilaginosa* has been reported to be able grow at 30 °C but not at 37 °C. However, *R. mucilaginosa* isolates found in the snow in Antarctica were resistant to fluconazole and able to grow at 37 °C.

*Penicillium chrysogenum* was previously reported as an opportunistic fungus in immunocompromised patients, resulting in otomycoses, endophthalmitis, keratitis, endocarditis, cutaneous infections, and systemic infection [86]; it was also able to cause bloodstream infection in immunocompromised patients [87] and cerebral disease in healthy individuals [88]. *Penicillium* taxa recovered from the ornithogenic soil samples and rocks of from continental and Antarctic Peninsula [89,90,91] displayed different pathogenic potential to human in vitro.

Kang et al. [15] detected organochlorine pesticides (OCPs) at different concentrations in the snow of East Antarctica and suggested that the air masses that arrive at the sampling sites mainly originate from the Indian and Atlantic Oceans which travel over the Antarctic continent, suggesting that the OCPs were subjected to long-range atmospheric transport and were deposited in surface snow. In recent years, several studies have focused on fungal cross-resistance between agricultural pesticides and clinical antifungal drugs [92,93,94]. Opportunistic fungi are also ubiquitous in the environment and can come into contact with pesticides and antifungal agents used in crops [94]. Since OCPs have chemical structures that are similar to those of antifungal azoles, their presence in the snow in Antarctica may explain the detection of *D. hansenii*, *R. mucilacinosa*, and *P. chrysogenum* resistant to agricultural and clinical antifungal drugs.

## 5. Conclusions

Our results show that the winter seasonal snow in the Antarctic Peninsula contains a diverse fungal community dominated by cosmopolitan ubiquitous basidiomycetous yeasts species previously found in tropical, temperate, and polar ecosystems. These species were found to display specific adaptations for their successful survival and colonization in the Antarctic snow microhabitat. The high densities of these cosmopolitan fungi suggest that they may be present in the air masses arriving at the Antarctic Peninsula from outside Antarctica, and these air masses are then pushed down by snow precipitation and compacted at high concentrations in the snow each year. Additionally, we detected environmental fungal isolates resistant to agricultural and clinical antifungals, some of which were able to grow at 37 °C. Further studies will be needed to characterize the innate virulence potential of these fungi against humans and animals.

## Figures and Tables

**Table 1 microorganisms-07-00445-t001:** Fungi obtained from Antarctic snow segments and identified by sequence comparison with the BLASTn match with the NCBI GenBank database.

Region	Snow Segment	Density (CFU L^−1^)	UFMGCB ^a^	Top BLAST Search Results (GenBank Accession Number)	Query Cover (%)	Identity (%)	N° of bp Analyzed	Proposed Taxa (GenBank acc. n°)
Deception Island (62° 58′ 55.6″ S; 60° 33′ 11.5″ W)	1	>300	18371	*Leucosporidium fragarium* (NG058330) ^e^	100	99	831	*Leucosporidium fragarium* (MN0655470) ^j^
		36.66	12598	*Penicillium cavernicola* (MH862709) ^b^	100	100	466	*Penicillium* sp. 1 (MK981319) ^g^
		10	12377	*Mortierella elongatula* (MH859811) ^b^	100	96	483	*Mortierella* sp. (MK889363) ^g^
		3.3	18377	*Rhodotorula mucilaginosa* (MH636067) ^e^	100	99	495	*Rhodotorula mucilaginosa* (MN0655160) ^j^
	2	>300	18378	*Phenoliferia glacialis* (LC203727) ^e^	100	100	815	*Phenoliferia glacialis* (MN065548) ^j^
		9.9	12592	*Cladosporium phyllactiniicola* (MH863929) ^b^	100	100	432	*Cladosporium* sp. 1 (MK981325) ^g^
		6	12376	*Antarctomyces**psychrotrophicus* (MH874317) ^b^	100	100	432	*Antarctomyces**psychrotrophicus* (MK889356) ^g^
		4	12597	*Thelebolus balaustiformis* (NR159056) ^b^	99	100	443	*Thelebolus balaustiformis* (MK889369) ^g^
		2	12375	*Penicillium cavernicola* (MH862709) ^b^ *Penicillium albocoremium* (KU896812) ^c^ *Penicillium solitum* (KU904363) ^d^	100 99 100	100 97 100	430 405 689	*Penicillium solitum* (MK981306) ^g^ (MN017378) ^h^ (MN056972) ^i^
	3	>300	18368	*Leucosporidium golubevii* (KY108283) ^e^	99	98	670	*Leucosporidium* sp. (MN065549) ^j^
		41.3	12371	*Leohumicola minima* (NR121307) ^b^	100	91	474	*Helotiales* sp. 1 (MK889360) ^g^
		4.6	12428	*Thelebolus balaustiformis* (NR159056) ^b^	100	100	487	*Thelebolus balaustiformis* (MK889370) ^g^
		5.3	12741	*Penicillium cavernicola* (MH862709) ^b^	100	99	468	*Penicillium* sp. 1 (MK981334) ^g^
		4	12372	*Penicillium rubens* (NR111815) ^b^ *Penicillium glycyrrhizacola* (KF021538) ^c^	100 99	100 98	464 398	*Penicillium* sp. 2 (MK981307) ^g^ (MN0173379) ^h^
		2	12380	*Penicillium tardochrysogenum* (MH865983) ^b^ *Penicillium chrysogenum* (AY495981) ^c^	100 100	99 97	470 377	*Penicillium**chrysogenum* (MK981309) ^g^ (MN045349) ^h^
		1	12373	*Penicillium roqueforti* (MH855127) ^b^ *Penicillium carneum* (AY674386) ^c^ *Penicillium psychrosexualis* (KU904362) ^d^	100 96 100	100 97 99	463 321 554	*Penicillium* cf. *psychrosexualis* (MK981308) ^g^ (MN045348) ^h^ (MN379931) ^i^
King George Island (62° 06′ 211″ S; 058° 27’ 627″ W)	1	>300	18388	*Phenoliferia psychrophenolica* (KY108774) ^e^	100	100	551	*Phenoliferia psychrophenolica* (MN065550) ^j^
		>300	18404	*Phenoliferia glacialis* (NG058369) ^e^	94	99	555	*Phenoliferia* cf. *glacialis* (MN065514) ^j^
		>300	18386	*Phaeococcomyces mexicanus* (NG058078) ^e^	100	98	490	*Phaeococcomyces* sp. (MN065551) ^j^
		>300	18400	*Rhodotorula mucilaginosa* (KY218728) ^e^	100	99	622	*Rhodotorula mucilaginosa* (MN065517) ^j^
		2.66	12430	*Penicillium tardochrysogenum* (MH865983) ^b^	100	100	476	*Penicillium* sp. 3 (MK981320) ^g^
		2.66	12388	*Thelebolus globosus* (MH862951) ^b^	100	99	499	*Thelebolus globosus* (MK889371) ^g^
		13.3	12391	*Alternaria multiformis* (MH862776) ^b^	100	100	447	*Alternaria multiformis* (MK889355) ^g^
		4.6	12395	*Penicillium rubens* (NR111815) ^b^ *Penicillium chrysogenum* (JF909937) ^d^	98 98	99 99	469 936	*Penicillium chrysogenum* (MK981310) ^g^ (MN379932) ^i^
		3.5	12390	*Neomicrosphaeropsis italica* (NR158247) ^b^	100	98	503	*Neomicrosphaeropsis* cf. *italica* (MK889364) ^g^
		2.6	12397	*Penicillium**tardochrysogenum* (MH865983) ^b^ *Penicillium chrysogenum* (JF909937) ^d^	100 100	99 100	467 920	*Penicillium chrysogenum* (MK981311) ^g^(MN379933) ^i^
		3.3	12389	*Ijuhya vitelina* (NR154100) ^b^	100	88	511	*Bionectriaceae* sp. 1 (MK889358) ^g^
		1.3	12393	*Thelebolus balaustiformis* (NR159056) ^b^	100	100	474	*Thelebolus balaustiformis* (MK889374) ^g^
		1.3	18399	*Mrakia gelida* (KY108585) ^e^	98	99	554	*Mrakia* sp. (MN065528) ^j^
	2	>300	18387	*Hamamotoa singularis* (KY107777) ^e^	100	99	435	*Hamamotoa singularis* (MN065527) ^j^
		>300	18483	*Rhodotorula mucilaginosa* (KY218730) ^e^	100	100	673	*Rhodotorula mucilaginosa* (MN065518) ^j^
		>300	18391	*Hamamotoa singularis* (KY107777) ^e^	100	97	551	*Hamamotoa* sp. (MN065526) ^j^
		>300	18478	*Vishniacozyma victoriae* (LC203739) ^e^	100	99	642	*Vishniacozyma victoriae* (MN0655520) ^j^
		88	18381	*Glaciozyma antarctica* (NG057664) ^e^	100	100	553	*Glaciozyma antarctica* (MN065525) ^j^
		7.3	12734	*Ijuhya vitelina* (NR154100) ^b^	100	87	453	*Bionectriaceae* sp. 1 (MK889359) ^g^
		6	12385	*Penicillium cavernicola* (MH862709) ^b^ *Penicillium glandicola* (KU896814) ^c^	100 93	100 82	465 407	*Penicillium* sp. 1 (MK981312) ^g^ (MN045350) ^h^
		1.3	12594	*Cladosporium phyllactiniicola* (MH863929) ^b^	100	100	465	*Cladosporium* sp. 1 (MK981333) ^g^
		1	12384	*Neomicrosphaeropsis italica* (NR158247) ^b^	100	98	482	*Neomicrosphaeropsis* cf. *italica* (MK889365) ^g^
	3	>300	18395	*Phenoliferia psychrophenolica* (KY108774) ^e^	100	100	565	*Phenoliferia psychrophenolica* (MN065553) ^j^
		>300	18393	*Rhodotorula mucilaginosa* (MF927654) ^e^	100	99	546	*Rhodotorula mucilaginosa* (MN065519) ^j^
		>300	18403	*Glaciozyma antarctica* (LC202043) ^e^	100	100	501	*Glaciozyma antarctica* (MN065524) ^j^
		>300	18493	*Phenoliferia glacialis* (LC203727) ^e^	100	100	590	*Phenoliferia glacialis* (MN0655540) ^j^
		8	12386	*Penicillium cavernicola* (MH862709) ^b^ *Penicillium albocoremium* (KU896812) ^c^	100 95	100 97	425 370	*Penicillium* sp. 1 (MK981313) ^g^ (MN045351) ^h^
		2.6	12735	*Penicillium caseifulvum* (MH862722) ^b^	100	100	373	*Penicillium* sp. 4 (MK981321) ^g^
		4	12387	*Penicillium commune* (NR111143) ^b^ *Penicillium palitans* (KU904360) ^d^	100 100	100 100	466 577	*Penicillium palitans* (MK981314) ^g^ (MN379934) ^i^
		4	12582	*Cladosporium**asperulatum* (MH863916) ^b^	100	100	475	*Cladosporium* sp. 2 (MK981326) ^g^
		4	12740	*Leohumicola minima* (NR121307) ^b^	100	91	482	*Helotiales* sp. 1 (MK889361) ^g^
Trinity Peninsula (63° 24’ 45.6″ S; 57° 00’ 164″ W)	1	>300	18481	*Phenoliferia glacialis* (LC203723) ^e^	100	99	678	*Phenoliferia glacialis* (MN065531) ^j^
		>300	18417	*Bannozyma**yamatoana* (AF189896) ^e^	100	99	563	*Bannozyma**yamatoana* (MN065523) ^j^
		>300	18489	*Vishniacozyma victoriae* (LC203739) ^e^	100	100	559	*Vishniacozyma victoriae* (MN065555) ^j^
		>300	18480	*Cystobasidium pallidum* (NG059006) ^e^	100	93	529	*Cystobasidium* sp. (MN065556) ^j^
		110	18410	*Phenoliferia psychrophenolica* (KY108774) ^e^	100	99	543	*Phenoliferia psychrophenolica* (MN065530) ^j^
		46	12405	*Penicillium caseifulvum* (MH862722) ^b^ *Penicillium longisporum* (KJ834467) ^c^	100 88	100 98	495 336	*Penicillium* sp. 4 (MK981316) ^g^ (MN075199) ^h^
		31.3	18408	*Dioszegia hungarica* (KY107643) ^e^	100	99	488	*Dioszegia hungarica* (MN065557) ^j^
		34	18423	*Dioszegia crocea* (KX096670) ^e^	100	100	448	*Dioszegia crocea* (MN065520) ^j^
		12.6	18411	*Genolevuria bromeliarum* (NG058291) ^e^ *Genolevuria bromeliarum* (NG058291) ^b^	98 61	94 95	542 860	*Genolevuria* sp. (MN065564) ^j^ (MN065510) ^g^
		6	12429	*Penicillium oregonense* (MH865652) ^b^ *Penicillium glandicola* (KU896814) ^c^	100 93	96 82	419 407	*Penicillium* sp. 5 (MK981315) ^g^ (MN045352) ^h^
		2	12404	*Thelebolus balaustiformis* (NR159056) ^b^	99	100	488	*Thelebolus balaustiformis* (MK889373) ^g^
	2	>300	18413	*Rhodotorula mucilaginosa* (KP223715) ^e^	100	100	596	*Rhodotorula mucilaginosa* (MN065512) ^j^
		>300	18425	*Phenoliferia psychrophenolica* (KY108774) ^e^	100	100	508	*Phenoliferia psychrophenolica* (MN065535) ^j^
		106	12427	*Penicillium tardochrysogenum* (NR1383081) ^b^ *Penicillium chrysogenum* (JF909937) ^c^ *Penicillium chrysogenum* (AY495981) ^d^	100 100 100	99 100 100	504 651 342	*Penicillium chrysogenum* (MK981317) ^g^ (MN075200) ^h^ (MN379935) ^i^
		8.6	18426	*Debaryomyces hansenii* (KX981201) ^e^	100	100	712	*Debaryomyces hansenii* (MN065522) ^j^
		2.6	18416	*Leucosporidium muscorum* (KY108280) ^e^ *Leucosporodium creatinivorum* (KC455908) ^b^	100 99	99 100	599 654	*Leucosporidium* sp. (MN121386) ^e^ (MN121387) ^g^
		1.3	18422	*Candida sake* (KY102375) ^e^	100	100	656	*Candida sake* (MN065558) ^j^
	3	>300	18414	*Phenoliferia glacialis* (LC203723) ^e^	100	99	653	*Phenoliferia* cf *glacialis* (MN0655320) ^j^
		>300	18415	*Phenoliferia psychrophenolica* (KY108774) ^e^	100	99	499	*Phenoliferia psychrophenolica* (MN065533) ^j^
		5.3	12402	*Penicillium tardochrysogenum* (MH865983) ^b^	100	99	419	*Penicillium* sp. 3 (MK981335) ^g^
		2.6	12732	*Cladosporium phyllactiniicola* (MH863929) ^b^	100	100	428	*Cladosporium* sp. 1 (MK981327) ^g^
Arctowski Peninsula (64 45 ’0″S; 62 25’ 0″ W)	1	>300	18430	*Phenoliferia psychrophenolica* (KY108774) ^e^	100	100	533	*Phenoliferia psychrophenolica* (MN065534) ^j^
		120	18428	*Genolevuria amylolytica* (NG057728) ^e^	100	99	455	*Genolevuria amylolytica* (MN065559) ^j^
		3.65	12432	*Antarctomyces**psychrotrophicus* (MH874317) ^b^	100	100	435	*Antarctomyces**psychrotrophicus* (MK889357) ^g^
		2	12433	*Cladosporium**phaenocomae* (MH865096) ^b^ *Cladosporium longisporum* (KJ834467) ^c^	100 94	100 97	435 355	*Cladosporium* sp. 3 (MK981324) ^g^ (MN075202) ^h^
		2	18436	*Debaryomyces hansenii* (LC219506) ^e^	100	100	546	*Debaryomyces hansenii* (MN065521) ^j^
	2	>300	18431	*Phenoliferia psychrophenolica* (KY108774) ^e^	100	100	571	*Phenoliferia psychrophenolica* (MN065536) ^j^
		>300	18477	*Phenoliferia glacialis* (LC203727) ^e^	100	100	510	*Phenoliferia glacialis* (MN065537) ^j^
		15.33	12431	*Cladosporium phyllactiniicola* (MH863929) ^b^	100	100	462	*Cladosporium* sp. 1 (MK981330) ^g^
		7.3	12408	*Cladosporium**variabile* (MH863132) ^b^	100	100	481	*Cladosporium* sp. 3 (MK981331) ^g^
		1.3	12587	*Penicillium caseifulvum* (MH862722) ^b^	100	100	488	*Penicillium* sp. 4 (MK981323) ^g^
		1	12407	*Thelebolus**balaustiformis* (NR159056) ^b^	98	100	439	*Thelebolus* cf. *balaustiformis* (MK889366) ^g^
Snow Island (62° 46’ 300″ S; 06° 15’ 240″ W)	1	>300	18455	*Phenoliferia psychrophenolica* (KY108774) ^e^	100	100	527	*Phenoliferia psychrophenolica* (MN065538) ^j^
		157.3	18439	*Mrakia gelida* (LC203698) ^e^	100	100	618	*Mrakia gelida* (MN065529) ^j^
		19.95	18441	*Phenoliferia glacialis* (LC203727) ^e^	100	100	631	*Phenoliferia glacialis* (MN065539) ^j^
		28	18475	*Holtermanniella wattica* (LC203693) ^e^	100	100	458	*Holtermanniella wattica* (MN065560) ^j^
		6.3	18443	*Rhodotorula mucilaginosa* (KY218730) ^e^	100	100	543	*Rhodotorula mucilaginosa* (MN065513) ^j^
		3.3	18491	*Genolevuria bromeliarum* (NG058291) ^e^	96	98	542	*Genolevuria* sp. (MN065563) ^j^
		1.3	18440	*Kabatiella bupleuri* (JN886792) ^e^ *Kabatiella bupleuri* (JN886792) ^b^	98 100	98 98	420 577	*Kabatiella* cf. *bupleuri* (MN065565) ^j^ (MN655511) ^g^
		2	12590	*Clathrosphaerina zalewskii* (MH856474) ^b^	100	91	423	*Hyaloscyphaceae* sp. (MK889362) ^g^
	2	>300	18445	*Phenoliferia psychrophenolica* (KY108774) ^e^	100	100	553	*Phenoliferia psychrophenolica* (MN065540) ^j^
		>300	18484	*Phenoliferia glacialis* (LC203727) ^e^	100	100	571	*Phenoliferia glacialis* (MN065561) ^j^
		0.6	18448	*Vishniacozyma victoriae* (LC203739) ^e^	100	99	599	*Vishniacozyma victoriae* (MN065562) ^j^
		2	12434	*Cladosporium**tenuissimum* (MH864840) ^b^	100	99	471	*Cladosporium* sp. 4 (MK981332) ^g^
		2	12411	*Penicillium tardochrysogenum* (NR1383081) ^b^ *Penicillium chrysogenum* (AY4959811) ^c^	100 100	100 99	523 416	*Penicillium chrysogenum* (MK981318) ^g^ (MN075201) ^h^
	3	>300	18437	*Phenoliferia glacialis* (LC203727) ^e^	100	100	571	*Phenoliferia glacialis* (MN065541) ^j^
		>300	18456	*Phenoliferia psychrophenolica* (FN550378) ^e^	100	99	514	*Phenoliferia psychrophenolica* (MN065542) ^j^
		6.65	12739	*Pseudogymnoascus destructans* (EU88492) ^b^	100	99	453	*Pseudogymnoascus destructans* (MK889375) ^g^
		2.66	12412	*Thelebolus globosus* (KX576510) ^b^	100	99	410	*Thelebolus globosus* (MK889367) ^g^
		2.6	12589	*Pseudogymnoascus verrucosus* (KJ755525) ^b^	100	100	400	*Pseudogymnoascus verrucosus* (MK889376) ^g^
		4	12588	*Pseudogymnoascus pannorum* (MH861038) ^b^	99	98	414	*Pseudogymnoascus* sp. (MK889378) ^g^
Robert Island (62 °37’ 941″ S; 059° 70’ 400″ W)	1	>300	18470	*Phenoliferia psychrophenolica* (FN550378) ^e^	100	100	525	*Phenoliferia psychrophenolica* (MN065543) ^j^
		>300	18461	*Phenoliferia glacialis* (KX773532) ^e^	100	100	506	*Phenoliferia glacialis* (MN065544) ^j^
		3.3	12420	*Cladosporium phyllactiniicola* (MH863929)^b^	98	100	430	*Cladosporium* sp. 1 (MK981328) ^g^
		4	12419	*Thelebolus**balaustiformis* (NR159056) ^b^	100	99	511	*Thelebolus* cf. *balaustiformis* (MK889368) ^g^
		1.3	12423	*Pseudocamarosporium africanum* (NR154294) ^b^ *Pseudocamarosporium africanum* (JX496368) ^c^	99 90	100 99	528 422	*Pseudocamarosporium africanum* (MK889379) ^g^ (MN075203) ^h^
	2	>300	18464	*Phenoliferia psychrophenolica* (KY108774) ^e^	100	99	538	*Phenoliferia psychrophenolica* (MN065545) ^j^
		2.66	12414	*Thelebolus balaustiformis* (NR159056) ^b^	100	100	425	*Thelebolus balaustiformis* (MK889372) ^g^
		1.33	12413	*Cladosporium phyllactiniicola* (MH863929) ^b^	100	100	469	*Cladosporium* sp. 1 (MK981329) ^g^
		1.3	12733	*Penicillium rubens* (NR111815) ^b^	98	100	434	*Penicillium* sp. 2 (MK981322) ^g^
	3	>300	18472	*Phenoliferia psychrophenolica* (KY108774) ^e^	100	100	528	*Phenoliferia psychrophenolica* (MN065546) ^j^
		>300	18490	*Rhodotorula mucilaginosa* (KY218730) ^e^	100	100	605	*Rhodotorula mucilaginosa* (MN065515) ^j^
		14	12416	*Antarctomyces pellizariae* (KX576510) ^b^ (KX790790) ^c^ (KY100007) ^d^	100	100	545	*Antarctomyces pellizariae* (KX576510) ^g^ (KX790790) ^h^ (KY100007) ^i^
		1	12438	*Pseudogymnoascus verrucosus* (KJ755525) ^b^	98	99	488	*Pseudogymnoascus* cf. *verrucosus* (MK889377) ^g^

^a^ UFMGCB, Culture of Microorganisms and Cells from the Federal University of Minas Gerais. Taxa subjected to BLAST analysis based on the ^b^ ITS, ^c^ β-tubulin, ^d^ Polymerase II, and ^e^ D1/D2 regions for the elucidation of taxonomic positions. ^f^ Taxonomic position suggested. Sequences of ^g^ ITS, ^h^ β-tubulin and/or ^i^ Polymerase II and ^j^ D1/D2 sequences deposited in GenBank database.

**Table 2 microorganisms-07-00445-t002:** Physicochemical parameters of melted snow sampled in different sites of Antarctic Peninsula and diversity indices fungal.

Parameters/Diversity Indices/Density	Sites Sampled					
	Deception Island	King George Island	Trinity Peninsula	Arctowski Peninsula	Snow Island	Robert Island
Conductivity (µS cm^−1^)	50	173	20	28	52	53
Resistivity (MΩ cm^−1^)	0.024	0.006	0.047	0.023	0,026	0.039
Total dissolved solids (ppm)	24	86	11	14	14.6	27
Oxidation Reduction Potential (mV)	190.9	310	256.8	255.9	361.2	230.2
pH	6.76	6.3	6.74	6.24	6.1	6.5
Salinity (ppt)	0.02	0.08	0.01	0.01	0.02	0.02
Number of taxa	16	31	21	11	20	14
Fisher α	2.69	4.63	3.09	1.71	3.17	2.06
Margalef’s	2.16	3.65	2.52	1.44	2.55	1.73
Simpson’s	0.74	0.92	0.90	0.74	0.84	0.84

**Table 3 microorganisms-07-00445-t003:** The minimal inhibitory concentration of benomyl, fluconazole, and amphotericin B against fungi isolated from seasonal snow of Antarctica.

Fungi	UFMGCB ^a^	MIC ^b^ at 25 °C		MIC ^b^ at 37 °C ^d^
		Benomyl	Fluconazole	Amphotericin B
*Debaryomyces hansenii*	18426	40	8	−
*D. hansenii*	18436	40	4	−
*Rhodotorula mucilaginosa*	18377	0.3125	64	0.03
*R. mucilaginosa*	18400	0.3125	64	0.03
*R. mucilaginosa*	18393	0.3125	64	0.03
*R. mucilaginosa*	18413	0.3125	64	0.03
*R. mucilaginosa*	18443	0.3125	64	0.125
*Penicillium chrysogenum*	12380	0	32	−
*P. chrysogenum*	12427	0	64	−
*Penicillium* sp. 3	12430	2.5	64	−
*Penicillium* sp. 2	12733	2.5	1	−
*Penicillium* sp. 4	12405	2.5	64	−

^a^ UFMGCB, Culture of Microorganisms and Cells from the Federal University of Minas Gerais. ^b^ MIC, Minimal inhibitory concentration at µg mL^−1^. ^d^ for selected fungi with MIC ≥ 64 µg mL^−1^ that were able to grow at 37 °C. 0, Absence of growth. −, Fungi incapable to grow at 37 °C.

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
