# Peer review of "Diversity, Distribution, and Ecology of Fungi in the Seasonal Snow of Antarctica"

_microorganisms, 2019, doi:10.3390/microorganisms7100445_

Round 1

Reviewer 1 Report

The paper entitled "Diversity, distribution, and ecology of fungi in the seasonal snow of Antarctica" presented by Menezes et al., is well structured and clearly described from methodology and results point of view.

Some minor recommendations as follow: 

line 45 - 47 - Industrial .... I think that this paragraph is not necessary in the introduction section. line 58 - please insert a reference at the end of the sentence. line 95 - please insert the sequences of the primers.

Author Response

Reviewer 1

The paper entitled "Diversity, distribution, and ecology of fungi in the seasonal snow of Antarctica" presented by Menezes et al., is well structured and clearly described from methodology and results point of view.

Some minor recommendations as follow:

Line 45 - 47 - Industrial .... I think that this paragraph is not necessary in the introduction section.

Answer: we understood the concern of the Reviewer; but we consider that the phrase is important to connect the industrial activities with the pollutants in the snow. However, to become the paragraph better as requested by the Reviewer, we changed it in the manuscript.

Line 58 - please insert a reference at the end of the sentence.

Anwer: we included the reference requested.

Line 95 - please insert the sequences of the primers.

Answer: the editor requested to decrease the protocols to avoid plagiarism. For this reason, we deleted some detailed informations in the Methods section, including the primers sequences.

Reviewer 2 Report

This manuscript identifies fungi from snow communities at different depths and sites across Antarctica through culturing and sequencing. Taxa were identified using BLAST, and cultured in the presence of antifungal compounds and at high temperatures (37C) to test for antifungal resistance and the potential to infect warm-blooded animals (virulence).  A range of taxa were recovered, but basidiomycete yeasts and globally widespread taxa dominated. A limited number of taxa were able to survive in the presence of high temperatures (1) or antifungal compounds (5). This paper represents a substantial amount of work and characterizes the diversity of an often overlooked habitat.  I think the paper could flow a little better in parts (mentioned below in specific comments), and would benefit from further discussing the degree of taxonomic similarity across sites and layers of snow.  I have provided some more specific comments below that are meant to strengthen the manuscript.  Finally, some of these details may have been discussed in the SI, but I was unable to access a Supplementary Document.

Specific Comments:

L15-16: This is a rather abrupt transition to growth in presence of antifungals and at high temps – I wonder if this might be smoothed out some if the motivation for testing for antifungal-resistant/high temp fungi in these habitats is discussed earlier.

L57-58: “These chemicals are then transported by winds to the Antarctic atmosphere, to be then deposited in the snow by precipitation.” Please provide a reference for this statement (perhaps the Kang et al. reference?)

L71: I am not finding a Supplemental document

L71: Was a uniform circumference used across cores, and the same depth for each segment (top, middle, bottom) used within cores and across cores?  Please clarify (this may already be in the SI, but I do not have the SI document).

L74: thaw -> thawed

L86-87: I assume it was the cultures that were sequenced (instead of a shotgun sequencing of what occurred on the membrane), but please clarify.

Table 1: Is snow segment 1 the top or the bottom?  How thick was snow (the M&M states that the bottom 10cm were used

Just before Table 2: “We did not detect any correlation between the fungal diversity indices and the physicochemical properties of seasonal Antarctic snow.” Please explain how this was tested.

Table 2: These values were obtained for all layers of the core, or the top layer?  Please clarify.  Also, please include how conductivity, resistivity, total dissolved solids, oxidation reduction potential, pH and salinity were obtained the M&M (I think these measures are first introduced here).

Table 2: Were there differences across layers for diversity measures?  These seem to be for the pooled core.  For instance, it would be interesting to know whether there is turnover across the same snow core, and if certain taxa are restricted to the bottom layer vs top layer.

How much similarity was there across sites?

Discussion p 2: an underscore has been retained following V. victoriae

Discussion: I would add the paragraph that begins this way to the end of the first paragraph in the Discussion “Generally, in extreme environments fungal communities are dominated by Ascomycota”

Discussion: add 37C to second subtitle in Discussion, as this is also discussed in this section.

Author Response

Revewer 2

This manuscript identifies fungi from snow communities at different depths and sites across Antarctica through culturing and sequencing. Taxa were identified using BLAST, and cultured in the presence of antifungal compounds and at high temperatures (37°C) to test for antifungal resistance and the potential to infect warm-blooded animals (virulence). A range of taxa were recovered, but basidiomycete yeasts and globally widespread taxa dominated. A limited number of taxa were able to survive in the presence of high temperatures (1) or antifungal compounds (5). This paper represents a substantial amount of work and characterizes the diversity of an often overlooked habitat. I think the paper could flow a little better in parts (mentioned below in specific comments), and would benefit from further discussing the degree of taxonomic similarity across sites and layers of snow. I have provided some more specific comments below that are meant to strengthen the manuscript. Finally, some of these details may have been discussed in the SI, but I was unable to access a Supplementary Document.

Specific Comments:

L15-16: This is a rather abrupt transition to growth in presence of antifungals and at high temps – I wonder if this might be smoothed out some if the motivation for testing for antifungal-resistant/high temp fungi in these habitats is discussed earlier.

Answer: a phrase was included to become the information more smoothed as requested by the Reviewer.

L57-58: “These chemicals are then transported by winds to the Antarctic atmosphere, to be then deposited in the snow by precipitation.” Please provide a reference for this statement (perhaps the Kang et al. reference?)

Answer: the reference was include. It is Kang et al. [16].

L71: I am not finding a Supplemental document

Answer: we submitted all supplementary material again and requested to editor become it available to the Reviewers.

L71: Was a uniform circumference used across cores, and the same depth for each segment (top, middle, bottom) used within cores and across cores? Please clarify (this may already be in the SI, but I do not have the SI document).

Answer: we included a new supplementary figure that show the top, middle, and base segments of the snow sampled.

L74: thaw -> thawed

Answer: the word was corrected.

L86-87: I assume it was the cultures that were sequenced (instead of a shotgun sequencing of what occurred on the membrane), but please clarify.

Answer: the Reviewer is correct. We included in the lines reported that we studied the cultivable fungi in the present study.

Table 1: Is snow segment 1 the top or the bottom?  How thick was snow (the M&M states that the bottom 10cm were used

Answer: the segment 1 is the top. We included a new supplementary figure that explain as snow segments were sampled.

Just before Table 2: “We did not detect any correlation between the fungal diversity indices and the physicochemical properties of seasonal Antarctic snow.” Please explain how this was tested.

Answer: the phrase was deleted.

Table 2: These values were obtained for all layers of the core, or the top layer? Please clarify. Also, please include how conductivity, resistivity, total dissolved solids, oxidation reduction potential, pH and salinity were obtained the M&M (I think these measures are first introduced here).

Answer: the physicochemical values were obtained from the mix of the all segments. As the parameters were obtained are included in the Methods.

Table 2: Were there differences across layers for diversity measures? These seem to be for the pooled core. For instance, it would be interesting to know whether there is turnover across the same snow core, and if certain taxa are restricted to the bottom layer vs top layer.

Answer: the diversity values were obtained from the mix of the all segments.

How much similarity was there across sites?

Answer: the similarities among the sites are represented in the Supplementary Figure 1, 2 and 3.

Discussion p 2: an underscore has been retained following V. victoriae

Answer. The underscore was removed.

Discussion: I would add the paragraph that begins this way to the end of the first paragraph in the Discussion “Generally, in extreme environments fungal communities are dominated by Ascomycota”

Answer: the phrase was included as requested by the Reviewer.

Discussion: add 37C to second subtitle in Discussion, as this is also discussed in this section.

Answer: the information was included.